# SOX2OT Long Noncoding RNA Is Regulated by the UPR in Oestrogen Receptor-Positive Breast Cancer

**Carole Ferraro-Peyret** [1,2,3,*], **Marjan E. Askarian-Amiri** [1,4], **Debina Sarkar** [1,4,5], **Wayne R. Joseph** [1], **Herah Hansji** [1,4,6], **Bruce C. Baguley** [1] **and Euphemia Y. Leung** [1,4,*]

1   Auckland Cancer Society Research Centre, University of Auckland, 85 Park Rd, Grafton, Auckland 1023, New Zealand; marjan.askarianamiri@gmail.com (M.E.A.-A.); debina.sarkar@otago.ac.nz (D.S.); w.joseph@auckland.ac.nz (W.R.J.); herah.hansji@mcri.edu.au (H.H.); b.baguley@auckland.ac.nz (B.C.B.)
2   Univ Lyon, ISPB Faculty of Pharmacy, INSERM 1052, CNRS5286, Cancer Research Centre of Lyon, F-69008 Lyon, France
3   Hospices Civils de Lyon, Biopathology of Tumours, GHE Hospital, F-69500 Bron, France
4   Molecular Medicine and Pathology Department, University of Auckland, 85 Park Rd, Grafton, Auckland 1023, New Zealand
5   Biochemistry Department, University of Otago, 710 Cumberland Street, Dunedin North, Dunedin 9016, New Zealand
6   Murdoch Children's Research Institute, The Royal Children's Hospital, Flemington Rd, Parkville, VIC 3052, Australia
*   Correspondence: carole.ferraro-peyret@univ-lyon1.fr (C.F.-P.); e.leung@auckland.ac.nz (E.Y.L.)

**Abstract:** Endoplasmic reticulum (ENR) stress perturbs cell homeostasis and induces the unfolded protein response (UPR). In breast cancer, this process is activated by oestrogen deprivation and is associated with tamoxifen resistance. We present evidence that the transcription factor SOX2 and the long noncoding RNA *SOX2* overlapping transcript (*SOX2OT*) are upregulated in oestrogen receptor-positive (ER+) breast cancer and in response to oestrogen deprivation. We examined the effect of the UPR on SOX2 and *SOX2OT* expression and the effect of *SOX2OT* on UPR pathways in breast cancer cell lines. The induction of the UPR by thapsigargin or glucose deprivation upregulates *SOX2OT* expression. This upregulation is also shown with the anti-oestrogen 4OH-tamoxifen and mTOR inhibitor everolimus in ER + breast cancer cells that are sensitive to oestrogen deprivation or everolimus treatment. *SOX2OT* overexpression decreased BiP and PERK expression. This effect of *SOX2OT* overexpression was confirmed on BiP and PERK pathway by q-PCR. Our results show that a long noncoding RNA regulates the UPR and evince a new function of *SOX2OT* as a participant of ENR stress reprogramming of breast cancer cells.

**Keywords:** breast cancer; LncRNA; SOX2OT; SOX2; UPR; 4-OH tamoxifen

## 1. Introduction

Despite extensive studies in breast cancer and more detailed knowledge of its molecular pathways, many aspects of breast cancer cell regulation are still enigmatic. Recent genomic and transcriptomic analyses showed that most of the transcripts are long noncoding RNAs (lncRNAs), and evidence for regulatory roles for lncRNAs continues to rise. The dysregulation of several lncRNAs in breast cancer has been reported [1–3] and the functions of this class of transcript require elucidation. Within an intronic region of the gene specifying the *SOX2* overlapping transcript (*SOX2OT*) lncRNA lies the *SOX2* gene, one of the main regulators of pluripotency (Figure 1, [4]). As described previously [4], *SOX2* and *SOX2OT* are differentially expressed in ER+ and ER- breast cancer. They are both upregulated in suspension culture under conditions that prioritise spheroid formation. Hence, we suggest that in breast cancer, *SOX2OT* is key to the regulation of *SOX2* expression. However, the mechanism of action of *SOX2OT* in breast cancer remains to be

fully defined. Expression analysis of murine *Sox2* and *Sox2OT* in different developmental systems has also elucidated the dynamically changing expression patterns of these two RNA species and has suggested important roles for these genes in normal development [5].

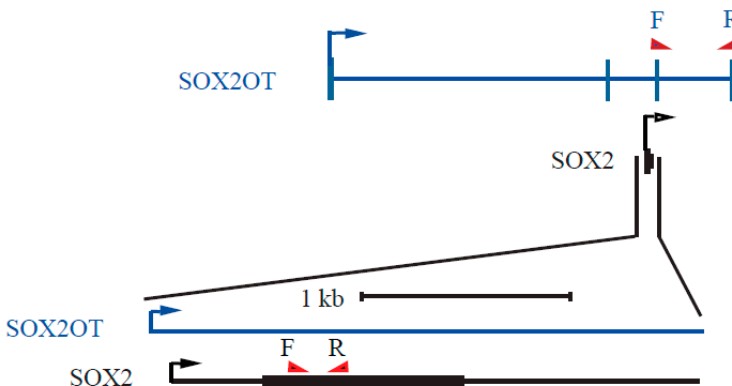

**Figure 1.** Schematic of the *SOX2* and *SOXOT* genes. The *SOX2* gene lies in an intronic region of the *SOX2 Overlapping Transcript* gene (*SOX2OT*). The direction of the transcription is shown with arrows. The triangles above each gene show the forward (**F**) and reverse (**R**) primers used in PCR. The *SOX2* gene is enlarged.

The dysregulation of *SOX2OT* and *SOX2* expression has also been shown in cancers such as glioma and kidney carcinoma [5], and significant correlations in expression of these two genes were found in breast cancer [4], as well as in oesophageal [6] and lung squamous cell carcinoma [7–9]. We have previously shown in breast cancer that the differential expression of *SOX2* and *SOX2OT* is oestrogen receptor dependent. *SOX2OT* and *SOX2* are more highly expressed in oestrogen receptor-positive (ER+) than in ER-negative (ER-) breast cancer cell lines [4]. Interestingly, high expression levels of *SOX2OT* and *SOX2* are associated with the sensitivity of breast cancer cells to tamoxifen [4]. To investigate the mechanism of action of *SOX2OT* in breast cancer further, we have examined the role of the unfolded protein response (UPR) pathway in the expression of this lncRNA.

Multiple mechanisms have been described for tamoxifen resistance to breast cancer. One of those pathways is known as the unfolded protein response (UPR). Three endoplasmic reticulum (ENR) stress transducers defining three distinct axes of the UPR have been identified so far and characterised as components of the UPR activation pathway (Figure 2). IRE1, PERK, and ATF6 are the three transmembrane inducers of ENR stress [10]. These three regulators of UPR are controlled by the ENR chaperone BiP, constitutively bound to them but dissociated under ENR stress. The UPR transiently inhibits protein synthesis and induces the production of chaperone molecules in order to restore ENR homeostasis and promote cell survival [11]. The failure of this rescue mechanism results in apoptotic cell death [12].

UPR activation is associated with poor prognosis in breast cancer [13,14]. Breast cancer cells of all subtypes have elevated UPR signalling with elevated BiP expression [15,16]. The UPR may favour oestrogen-dependent breast cancer survival when oestrogen availability is low [17–19]. The IRE-1 pathway is activated in ER+ breast cancer: it induces the splicing of XBP1 mRNA, consequently increasing the abundance of its pro-survival target XBP1s, which are strongly correlated with ER alpha expression in breast cancer [14,20]. It has been shown that silencing of XBP1, or inhibition of IRE1 by the pharmacological inhibitor STF-80310 or MKC866, reverses resistance to anti-oestrogen therapies [21–23]. XBP1s expression promotes the survival of triple-negative breast cancer (TNBC) cells [24]. Finally, the PERK pathway is activated in breast cancer cells resistant to tamoxifen [20,21,25].

The ability of the UPR to regulate gene expression and protein synthesis and its contribution to oncogenesis have been well documented. Recent results have shown that the UPR can regulate post-transcriptional networks either by *regulated IRE1-dependent decay* (RIDD) of selected mRNAs [26] or by modulation of expression of micro-RNAs [27]. The

UPR can also suppress the expression of several miRNAs that regulate either the PERK or the IRE1 pathway. However, an added layer of regulatory complexity that entails the role of lncRNA in the UPR pathway has been barely explored. Here, we investigate the gene expression pattern of *SOX2OT* in a UPR-induced system. Our results show that *SOX2OT* expression can be upregulated by ENR stress inducers and that this lncRNA can also downregulate the PERK pathway and BiP expression in ER+ breast cancer cell lines.

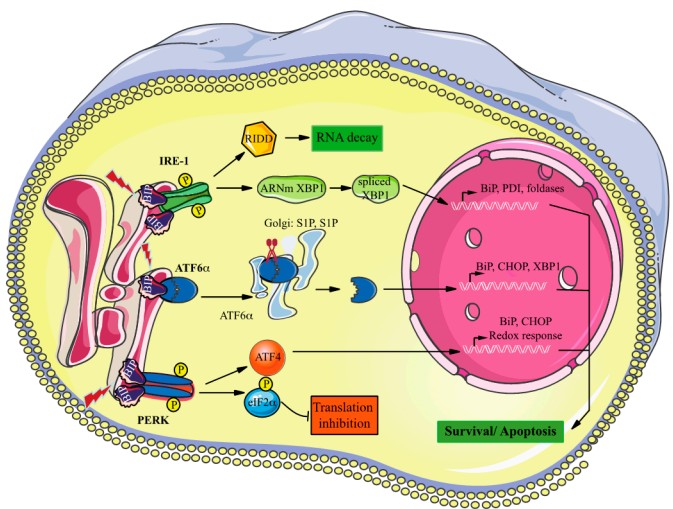

**Figure 2.** Schematic of pathways involved in the UPR.

## 2. Material and Methods

### 2.1. Cell Lines

As previously described, the MCF-7 and MDA-MB-231 cell lines were purchased from the ATCC, grown in alpha-MEM containing 5% foetal calf serum (FCS), penicillin/streptomycin (100 U/mL and 100 μg/mL, respectively), and insulin/transferrin/selenium supplement (Roche). Long-term oestrogen-deprived sublines TamC3 and TamC6 were generated by growing the MCF-7 cells in oestrogen-deprived medium (phenol red-free RPMI 1640 (GIBCO) with 5% charcoal-stripped foetal bovine serum) and penicillin/streptomycin (10 U/mL and 10 μg/mL), for 10 months, [28–31]. The T47D cell line was purchased from the ATCC, grown in alpha-MEM containing 5% foetal calf serum (FCS) and penicillin/streptomycin (100 U/mL and 100 μg/mL).

For glucose depletion, cells were seeded in six-well plates and maintained 48 h under normal culture conditions, i.e., 25 mM glucose concentration, 5% FCS, 5% $CO_2$, 21% $O_2$. The medium was replaced with no glucose DMEM (GIBCO) containing 5% FCS, penicillin/streptomycin (100 U/mL and 100 μg/mL), and insulin/transferrin/selenium supplement (Roche).

### 2.2. Reagents

Thapsigargin was obtained from Applichem (St. Louis, MO, USA), Everolimus from Selleck Chemicals (Houston, TX, USA), and 4-OH Tamoxifen is from Sigma-Aldrich (St. Louis, MO, USA). Anti-PERK (No. 3192) antibody was purchased from Cell Signaling Technology (Beverly, MA, USA). Anti-BIP antibody (610978) was from BD Laboratories™ (Franklin Lakes, NJ, USA). Antibody against α-tubulin was from Sigma-Aldrich (St. Louis, MO, USA).

### 2.3. Protein Extraction and Western Blot Analysis

As described in detail previously [29], breast cancer cell lines were grown to log-phase, washed twice with ice-cold PBS, and lysed in SDS lysis buffer according to the manufacturer's protocol (Cell Signalling Technology, Danvers, MA, USA). Protein concentration was quantified using the bicinchoninic acid reagent (Sigma). Cell lysates containing 20 μg

of protein were separated by electrophoresis on a 4–10% SDS-PAGE gel (Life Technologies) and transferred to a polyvinylidenedifluoride membrane (PVDF) (Millipore, Billerica, MA USA). Blocking of nonspecific binding was achieved in a 0.1% Tween 20 Tris-buffered saline solution containing 5% *w/v* non-fat dry milk powder for 1 h. Membranes were incubated with primary antibodies overnight at 4 °C, washed, and incubated with the corresponding immunoperoxidase-conjugated secondary antibody (Santa Cruz Biotechnology) for 1 h at room temperature. Bound antibody was visualised using SuperSignal West Pico (Thermo Scientific, Waltham, MA, USA) or ECL Select (Amersham) and the chemiluminescence detection system by Fujifilm Las-3000.

### 2.4. Reverse Transcription, cDNA Synthesis, and Quantitative PCR and PCR

As described in detail previously [4], oligo-dT and random primers were used to reverse transcribe RNA with qScript Flex cDNA kit (Dnature) according to the manufacturer's instructions. For RT-qPCR analysis, qPCR was performed using gene-specific primers (Supplementary Materials Table S1) and Sybr Green MasterMix (Life Technologies), and expression values normalised relative to *GAPDH* and *HPRT* mRNA expression.

### 2.5. Ectopic Expression of SOX2 and SOX2OT

This has been described in detail previously [4]. Constructs overexpressing *SOX2* (NM_003106) and *SOX2OT* (NR_004053) and control empty plasmid (vector), Ex-NEG-M95, and EX-hLUC-M90, respectively, were purchased from GeneCopoeia. Both plasmids express SV40-mCherry-IRES-puromycin resistance, allowing detection of transfected cells. Breast cancer cells (MDA-MB-231) were chosen because of the low expression of SOX2OT and SOX2 as we have previously shown [4]. The cells were transfected with 5 µg of DNA and Lipofectamine Plus (Invitrogen) according to the manufacturer's instructions. Three biological replicates for each construct were made, the transfected cells treated with puromycin and selected on the basis of mCherry expression by fluorescence-activated cell sorting (FACS), as previously described [4]. The sorted cells were maintained in the presence of puromycin.

### 2.6. Statistical Analysis

Results are presented as mean $\pm$ SEM. As described in detail previously [4], t-tests or Mann–Whitney rank sum tests were used for comparison between two groups. Correlation analysis was performed with Spearman's rank correlation coefficient (R) and statistical significance (*p*) using SigmaPlot. $p < 0.05$ (*), $p < 0.01$ (**) or $p < 0.001$ (***) were indications of statistical significance.

## 3. Results

### 3.1. Relationship between Expression of SOX2OT and ENR STRESS Stress-Inducible Genes in Breast Cancer

*SOX2OT* and *SOX2* transcripts are upregulated in tamoxifen-resistant cell lines [4,32]. The activation of the UPR in ER+ breast cancer cell lines following tamoxifen treatment has also been shown previously [33–35]. We initially analysed the genome wide-RNA transcript profile of breast cancer samples from the Cancer Genome Atlas (TCGA) by RNAseq dataset (TGCA8BRCA_exp_HiSeqV2-2015-02-24-160222) including 1025 samples from breast cancer patients. Interestingly, we found different patterns of expression of UPR-inducible genes relative to *SOX2OT* but not *SOX2* (Figure 3). As shown in Table 1, the expression of *XBP1* and *SOX2OT* were positively correlated (Spearman's correlation coefficient $r = 0.355$, $p = 2.0 \times 10^{-7}$). This analysis showed no correlation between the expression of *BiP* and *SOX2OT* while the expression of the other genes examined was negatively correlated with that of *SOX2OT,* with a weak correlation. However, we found no correlation of ER stress-inducible genes with *SOX2* with the exception of *PDIA4* ($r = 0.92$, $p = 1.4 \times 10^{-3}$).

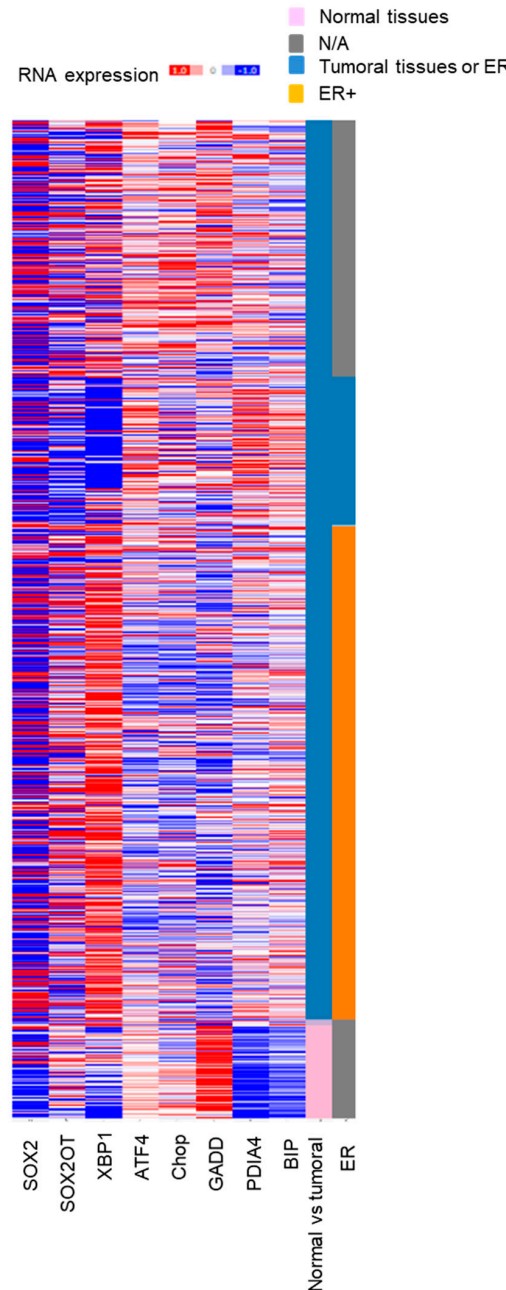

**Figure 3.** Heat map showing the expression of *SOX2*, *SOX2OT*, and UPR-related genes. It shows the expression of genes in the PERK pathway (*ATF4*, *CHOP/ddit3*, *GADD/pp1r15a*), ENR-associated degradation (*BiP/hspa5*, *PDIA4*), and the IRE1 pathway (*XBP1*) in breast cancer samples. Data were from TCGA invasive breast carcinoma (TGCA8BRCA_exp_ HiSeqV2-2015-02-24-160222). The samples were classified based on their receptor status (oestrogen receptor (ER), HER2, or progesterone receptor (PR)).

**Table 1.** Spearman correlation coefficients of relative gene expression for *SOX2OT* and *SOX2* compared to the expression of genes of the UPR pathways in TGCA samples (n = 1025).

|  | *ATF4* | *CHOP* | *GADD* | *PDIA4* | *XBP1* | *BiP* |
|---|---|---|---|---|---|---|
| *SOX2OT* | −0.195 | −0.167 | −0.148 | −0.171 | 0.355 | 0.019 |
| *p* value | $7.26 \times 10^{-12}$ | $4.95 \times 10^{-9}$ | $2.25 \times 10^{-7}$ | $1.92 \times 10^{-9}$ | $2.00 \times 10^{-7}$ | 0.508 |
| *SOX2* | 0.00381 | 0.0744 | −0.0549 | 0.0918 | 0.0158 | 0.0311 |
| *p* value | 0.894 | $9.48 \times 10^{-3}$ | 0.848 | $1.36 \times 10^{-3}$ | 0.581 | 0.278 |

### 3.2. SOX2OT Expression Is Upregulated in ER + Stressed Cells

To investigate whether lncRNA *SOX2OT* or *SOX2* is differentially expressed by ENR stress, two breast cancer cell lines (MCF-7 and T47D) were treated with the UPR inducer thapsigargin (Tg). We also studied the effect of two other drugs used to treat ER+ breast cancer cells: the anti-oestrogen 4OH-Tam and the mTOR inhibitor everolimus, both of which have been shown to induce the UPR in breast cancer cell lines [36] and in MCF-7 and T47D (Supplementary Materials Figure S1). Cells were assessed for PERK phosphorylation and BiP induction as markers of UPR activation using Western blot analysis (Supplementary Data S1). After 16h of treatment with Tg, a significant upregulation of *SOX2OT* expression was detected in both cell lines ($9.3 \pm 2.26$ fold in MCF-7 cells and $3.8 \pm 0.96$ fold in T47D cells) (Figure 4). This induction was also observed for *SOX2* expression in both cell lines. Treatment with 4OH-Tam and everolimus also upregulated *SOX2OT* in MCF-7 cells, while everolimus significantly induced *SOX2OT* in T47D cells. The *SOX2* level was significantly upregulated in MCF-7 and T47D cells treated with Tg or 4OH-Tam but not with everolimus.

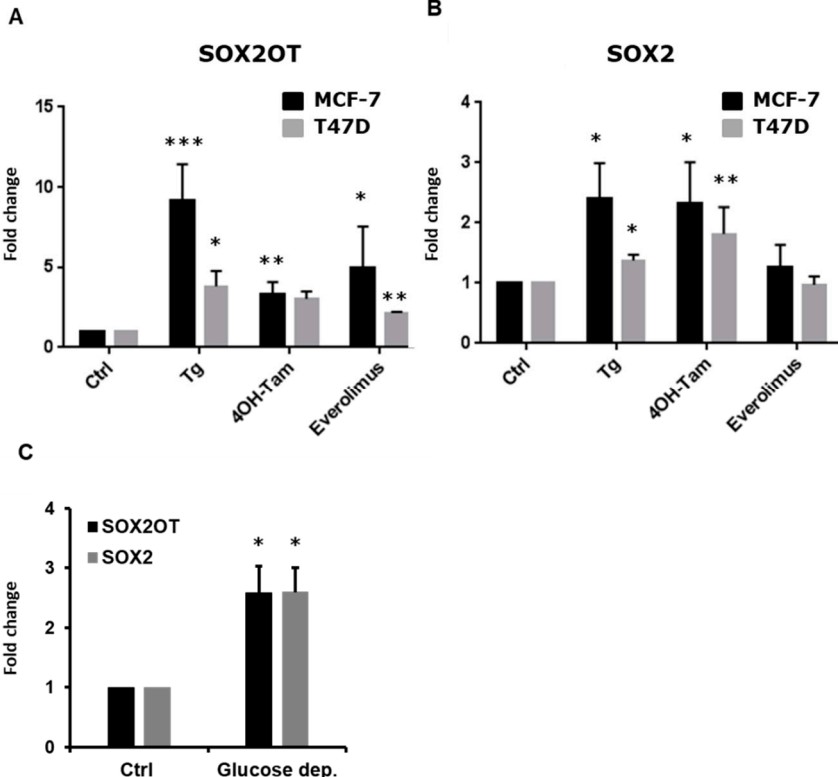

**Figure 4.** Expression of *SOX2OT* and *SOX2* following UPR induction in ER+ breast cancer cell lines. MCF-7 and T47D cells were incubated in a control medium (Ctrl) or in a medium containing the ER stress-inducing agents thapsigargin (Tg, 300 nM), 4OH-Tam (0.1 μM), or everolimus (10 nM) for 16 h. (**A,B**): The effects of Tg, 4OH-Tam, and everolimus on the expression of *SOX2OT* and *SOX2* were measured by RT-qPCR and are relative to the housekeeping genes *HPRT* and *GAPDH*. (**C**): MCF-7 cells were cultivated in media containing 25 mM glucose (Ctrl) or lacking glucose (Glucose dep) for 48h. The expression of *SOX2OT* and *SOX2* was measured by RT-qPCR and is relative to *HPRT* and *GAPDH*. Error bars represent standard deviations of at least 3 independent experiments. * $p < 0.05$, ** $p < 0.01$, *** $p < 0.001$.

MCF-7 cells were also grown under conditions of glucose deprivation, an intrinsic inducer of the UPR. After 48h of cultivation in glucose-depleted medium, significant increases in the expression of *SOX2OT* and *SOX2* were detected (Figure 4C).

As the UPR is upregulated in tamoxifen-resistant cells, we analysed the effects of Tg, 4OH-Tam, and everolimus in two oestrogen-deprived and -resistant breast cancer cell lines derived from MCF-7 cells: TamC3 and TamC6 [28–31]. We have previously shown

that the sensitivity to mTOR inhibition of these two cell lines is different: TamC6 cells are highly sensitive to everolimus, as compared to TamC3 cells [29–31]). Here, we show that Tg upregulated the expression of *SOX2OT* in both cell lines, while *SOX2* is upregulated only in the TamC6 cell line (Figure 5). 4OH-Tam did not induce the expression of these genes in TamC6 cells, whereas in TamC3 *SOX2OT* expression was increased (Figure 5). Everolimus, on the other hand, induced a significant increase of *SOX2OT* expression in TamC6 cells but not in TamC3 cells, while *SOX2* was upregulated by everolimus in both cell lines.

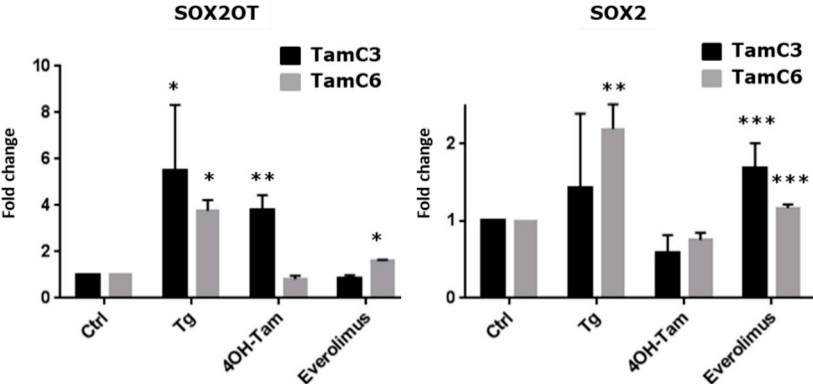

**Figure 5.** Expression of *SOX2OT* and *SOX2* following UPR induction in oestrogen-deprived cancer cells. TamC3 and TamC6 were incubated in a control medium (Ctrl) or in a medium containing the ENR stress-inducing agents thapsigargin (Tg, 300 nM), 4OH-Tam (0.1 μM), or everolimus (10 nM) for 16h. The expression of *SOX2OT* and *SOX2* was measured by RT-qPCR and is relative to that of *HPRT* and *GAPDH*. Error bars represent standard deviations of three independent experiments. * $p < 0.05$, ** $p < 0.01$, *** $p < 0.001$.

### 3.3. SOX2OT, but Not SOX2, Regulates BiP Expression and PERK Activation

To study further the role of the *SOX2OT* transcript in the UPR in breast cancer, we used the MDA-MB-231 cell line, which has low expression of *SOX2OT*, and in which we had induced ectopic expression of *SOX2OT* lncRNA [4]. The activation of the UPR pathway in cells overexpressing either *SOX2OT* or *SOX2* was studied at a transcriptional level, quantifying *BiP* and the PERK targets *ATF4* and *CHOP* by RT-qPCR. When *SOX2OT* was overexpressed, *BiP* and *CHOP* expression was significantly reduced, to 50% of control values (Figure 6A), while *XBP1s* mRNA expression was not significantly changed (Figure 6B). However, in SOX2 overexpressing cells *ATF4* was upregulated, while *CHOP* and *BiP* showed no significant change (Figure 6A). *XBP1s* mRNA was quantified by RT-qPCR and was not modified by *SOX2* overexpression (Figure 6B). We further investigated whether *SOX2OT* could inhibit BiP and the PERK by Western blot analysis for BiP and PERK expression. Our results show that the expression of BiP and PERK decreased in *SOX2OT* overexpressing cells (Figure 6C).

A positive correlation for the expression of *SOX2OT* and *SOX2* has been reported [4,5]. Therefore, we examined whether the downregulation of PERK and BiP was dependent on the SOX2 transcription factor or whether *SOX2OT* reduced the expression of these proteins independently of *SOX2*. In SOX2-overexpressing cells (Supplementary Materials Figure S2), when *BiP*, *CHOP*, and *ATF4* were quantified at the mRNA level by RT-qPCR, *ATF4* was slightly but significantly upregulated, compared to the control cells (Figure 6A) (Fold change = 1.32, $p < 0.001$).

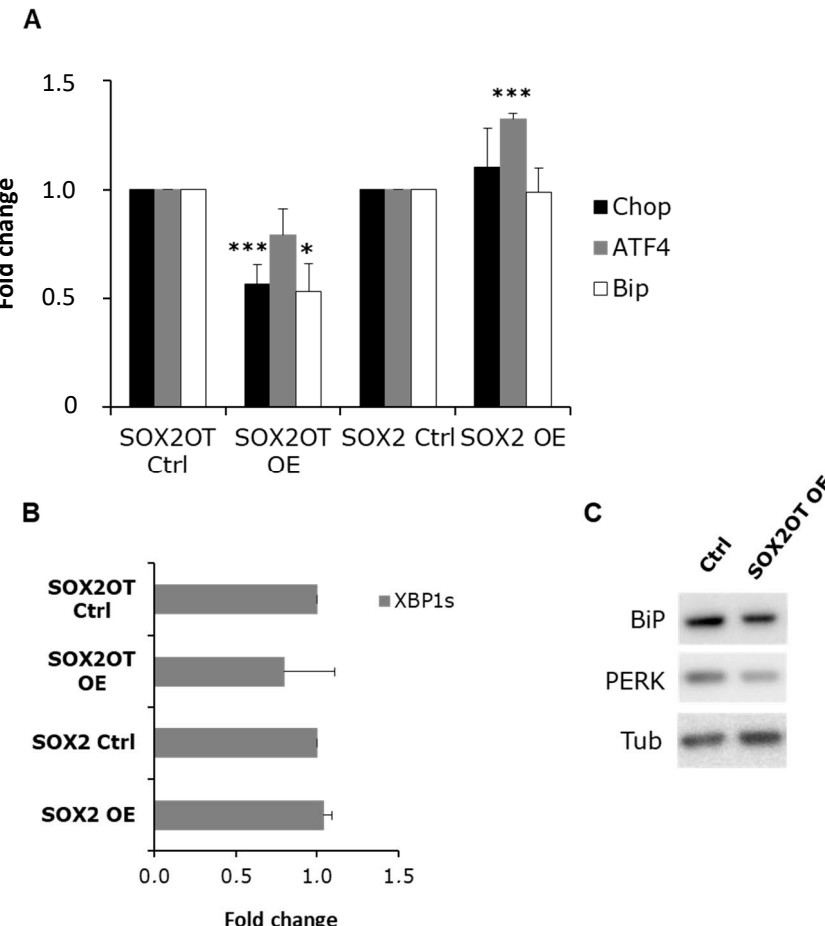

**Figure 6.** Effect of ectopic expression of *SOX2OT* or *SOX2* on the UPR signalling pathway. MDA-MB-231 cells were transfected with plasmids containing the *SOX2OT* gene (NR_004053.3) or the *SOX2* gene (NM_003106), or with their respective control vectors. (**A**): Expression of *BiP, ATF4*, and *CHOP* relative to that of *HPRT* and *GAPDH* was measured by RT-qPCR. (**B**): Relative expression of *XBP1s* was measured by RT-qPCR relative to that of *HPRT* and *GAPDH*. (**C**): Protein expression levels of BiP and PERK were examined using Western Blot analysis. α-tubulin was used as loading control. Error bars represent standard deviations of three independent clones. * $p < 0.05$, *** $p < 0.001$.

## 4. Discussion

Since the discovery of the adaptive response to the disruption of endoplasmic reticulum homeostasis, the UPR has emerged as having a major role in modulating the expression of cancer-related genes, notably through transcriptional or post-transcriptional changes. Here, we describe a new layer of regulatory mechanisms of the UPR, affected by the lncRNA *SOX2OT*.

Thousands of lncRNAs have been identified in cancer cells [37], but very little is known about their functions and mechanisms of action. Their localisation in varying subcellular compartments (nucleus or cytoplasm), and their ability to bind to a variety of targets suggest that they should regulate gene expression and also have an effect on post-transcriptional regulation or structural interaction. *SOX2OT* was first described in 2009 [3], and little is known about its role and the regulation of its expression. *SOX2OT* is expressed in different cancers, including breast [4,38,39], oesophagus [6], and lung [7,9]. Its effect on proliferation depends on the cell line studied. In lung cancer, cell lines *SOX2OT* inhibits cell cycle progression by regulating the expression of the histone–lysine N-methyltransferase enzyme EZH2 [7]. Conversely, in breast cancer cell lines, *SOX2OT* overexpression reduced proliferation and increased anchorage-independent growth [4].

In this study, we have shown that two ER-stress inducers —the decrease of glucose availability or the inhibition of the endoplasmic reticulum Ca$^{2+}$-ATPase by Tg— induced the expression of *SOX2OT.* Everolimus and 4OH-Tam, two drugs that are used to treat breast cancer and are known to induce the UPR in breast cancer cell lines [21,40], also induced an upregulation of the *SOX2OT* transcript, suggesting that *SOX2OT* and the UPR are related. This was confirmed using the MDA-MB-231 TNBC (triple-negative breast cancer) cell line that has a very low expression of *SOX2OT* and a strong expression of BiP. Here, we have shown that the overexpression of *SOX2OT* downregulates BiP expression and PERK pathway activity.

*SOX2* lies in an intron of the *SOX2OT* gene and is positively correlated with *SOX2OT* expression in breast cancer [4]. Therefore, *SOX2OT* is proposed to contribute to the transcriptional regulation of *SOX2* [6,7]. We observed that when MCF-7 or T47D breast cancer cells were treated with 4OH-Tam, *SOX2* expression and *SOX2OT* expression increased concordantly. However, *SOX2* overexpression neither decreased BiP expression nor affected the PERK pathway. These results show that the effect of *SOX2OT* on BiP or PERK pathway (measured by *ATF4* and *CHOP* RNA expression level) is independent of SOX2. Thus, *SOX2OT* has other targets yet to be identified. Furthermore, when the UPR was activated by Tg or glucose depletion, we observed that the correlation between *SOX2* and *SOX2OT* abundance was lost, suggesting that other factors regulate *SOX2* expression. This dissociation of *SOX2OT* and *SOX2* expression is also observed in TNBCs that have downregulated *SOX2OT* but still expressed *SOX2* [4]. Feng et al. [41] have demonstrated that in TBNC, cells that are prone to EMT have a high level of expression of BiP. This high expression may arise in part from *SOX2OT* downregulation and may promote the selection of more aggressive phenotypes or stem cell differentiation.

Other lncRNA molecules are known to be regulated by the UPR including *MALAT1* [42]. Its splicing is enhanced by PERK during infection by flavivirus [41]. These results suggest that the activation of the PERK pathway enhances the translation of specific mRNAs or miRNA as well as the expression of lncRNAs. The analysis of TCGA breast cancer samples showed a positive correlation between *XBP1* and *SOX2OT,* notably, in HER2 negative breast cancer samples (Supplementary Materials Table S2), suggesting that the IRE1 pathway could also regulate lncRNA.

Numerous studies have demonstrated that oestrogen deprivation induced a UPR [33–35]. In ER+ breast cancer cell lines, ER alpha-targeted therapy increased aggregation of ER alpha in the cytoplasm and increased UPR signalling [43]. The activation of UPR contributes to the development of resistance to oestrogen deprivation, as demonstrated by the overexpression of *XBP1s* [23] and BiP [44,45] or the activation of the PERK pathway [20,21,25]. Since *SOX2OT* is expressed early during UPR activation, it can participate in the reprogramming of gene expression by the UPR during the acute phase of oestrogen deprivation-mediated stress and lead to the emergence of resistant phenotypes. When treated with 4OH-Tam, MCF-7, T47D or TamC3 cells increased the expression of *SOX2OT*, whereas, in TamC6 cells, 4OH-Tam has no effect on *SOX2OT* expression, suggesting that in some resistant cell lines, upon chronic induction of the UPR, the mechanisms of regulation of *SOX2OT* expression are lost. The same observation was made following treatment with the mTOR inhibitor everolimus that has no effect on *SOX2OT* expression when cells are resistant to mTOR inhibition.

Together, our results show that long noncoding RNA can be considered a regulator of the UPR and provide evidence for a new function of *SOX2OT* as a participant of ENR stress reprogramming of breast cancer cells. Whether this LncRNA mediates its function through interactions with proteins, RNA, DNA, or a combination of these remains to be investigated. Therefore, overcoming *SOX2OT* overexpression and its effect on the PERK pathway or on SOX2 expression could result in the adaptation of the cells to ENR stress and consequently in the resistance to anti-oestrogen treatment and/or the promotion of EMT (Figure 7). Further investigations are needed to determine whether *SOX2OT* may provide a basis for new therapeutics to modify UPR in breast cancer.

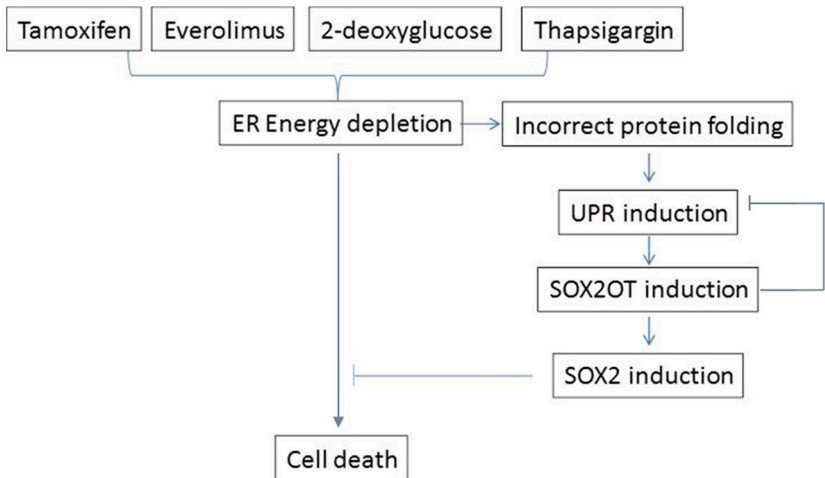

**Figure 7.** Hypothetical schema of *SOX2OT* lncRNA and UPR interrelation.

**Supplementary Materials:** The following are available online at https://www.mdpi.com/article/10
.3390/sci3020026/s1, Figure S1: Expression of *SOX2OT* lncRNA and *SOX2* following UPR induction
in ER+ breast cancer cell lines, Figure S2: Comparison of SOX2 expression in control and SOX2
overexpressing MDA-MB-231, Table S1: List of primers used in these experiments, Table S2: Spearman
correlation coefficients of relative gene expression for *SOX2OT* compared to the expression of genes
of the UPR pathways in TCGA Her2 neg/pos or Hormone neg/pos samples

**Author Contributions:** Conceptualization, C.F.-P., E.Y.L. and M.E.A.-A.; methodology, C.F.-P., D.S.
and W.R.J.; software, C.F.-P. and H.H.; validation, E.Y.L. and M.E.A.-A.; formal analysis, C.F.-P. and
D.S.; investigation, C.F.-P.; resources, E.Y.L. and M.E.A.-A.; data curation, C.F.-P.; writing—original
draft preparation, C.F.-P., E.Y.L., M.E.A.-A. and B.C.B.; writing—review and editing, C.F.-P., E.Y.L.;
visualization, all authors; supervision, B.C.B.; project administration, B.C.B..; funding acquisition, C.F.-
P., E.Y.L. and M.E.A.-A.; All authors have read and agreed to the published version of the manuscript.

**Funding:** C.F.-P. is supported by La Ligue Saône et Loire and the Hospices Civils de Lyon. M.E.A.-A.
is supported by the Genesis Oncology Trust fund. E.Y.L. is supported by the NZ Breast cancer
foundation.

**Institutional Review Board Statement:** Not applicable.

**Informed Consent Statement:** Not applicable.

**Data Availability Statement:** Data and materials are available on request from the corresponding
authors.

**Acknowledgments:** We thank G.J. Finlay for his careful reading of the paper and the constructive
comments he gave. C.F.-P. is supported by La Ligue Saône et Loire and the Hospices Civils de
Lyon. M.E.A.-A. is supported by the Genesis Oncology Trust. E.Y.L. is supported by the NZ Breast
Cancer Foundation.

**Conflicts of Interest:** The authors declare no conflict of interest.

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
