# Peer review of "SOX2OT Long Noncoding RNA Is Regulated by the UPR in Oestrogen Receptor-Positive Breast Cancer"

_sci, doi:10.3390/sci3020026_

Round 1

Reviewer 1 Report

the study only design overexpression of the related gene to explain the phenomena observe. it provide clue to the function. However, it would be more scientific sound to include gene knockdown or knockout data to support the conclusion. 

Author Response

the study only design overexpression of the related gene to explain the phenomena observe. it provide clue to the function. However, it would be more scientific sound to include gene knockdown or knockout data to support the conclusion.  The main conclusion in our manuscript, that expression of SOX2OT is correlated to expression of key components of UPR pathway, constitutes a novel finding. This conclusion is supported by the data linking SOX2OT overexpression to down-regulation of key components of UPR pathway, including BiP, Chop and ATF4, in the TCGA database. Our conclusion does not require data from gene knockout studies, although we agree that research in this direction would be useful in the future. We have modified the discussion to provide a clearer link between the results and the conclusion, and have also improved the introduction and modified the references in line with the reviewer’s comments.